

# A novel text sentiment analysis system using improved depthwise separable convolution neural networks

Xiaoyu Kong and Ke Zhang

Wuxi Vocational Institute of Commerce, Wuxi, Jiangsu, China

Corresponding author
Xiaoyu Kong,
kongxiaoyu@wxic.edu.cn

## ABSTRACT

Human behavior is greatly affected by emotions. Human behavior can be predicted by classifying emotions. Therefore, mining people's emotional tendencies from text is of great significance for predicting the behavior of target groups and making decisions. The good use of emotion classification technology can produce huge social and economic benefits. However, due to the rapid development of the Internet, the text information generated on the Internet increases rapidly at an unimaginable speed, which makes the previous method of manually classifying texts one-by-one more and more unable to meet the actual needs. In the subject of sentiment analysis, one of the most pressing problems is how to make better use of computer technology to extract emotional tendencies from text data in a way that is both more efficient and accurate. In the realm of text-based sentiment analysis, the currently available deep learning algorithms have two primary issues to contend with. The first is the high level of complexity involved in training the model, and the second is that the model does not take into account all of the aspects of language and does not make use of word vector information. This research employs an upgraded convolutional neural network (CNN) model as a response to these challenges. The goal of this model is to improve the downsides caused by the problems described above. First, the text separable convolution algorithm is used to perform hierarchical convolution on text features to achieve the refined extraction of word vector information and context information. Doing so avoids semantic confusion and reduces the complexity of convolutional networks. Secondly, the text separable convolution algorithm is applied to text sentiment analysis, and an improved CNN is further proposed. Compared with other models, the proposed model shows better performance in text-based sentiment analysis tasks. This study provides great value for text-based sentiment analysis tasks.

## INTRODUCTION

People's spiritual and material lives have been steadily improved as a result of the widespread dissemination of network information and the ongoing acceleration of technological development. Additionally, the relationship between societal needs and the entirety of the information age has evolved into one that is inseparable and interconnected. The birth of

new forms of media can be traced back to the fulfillment of social needs. As an important carrier of information dissemination, the Internet has made self-media platforms an important way for more and more people to express their opinions and express their emotions. At the same time, more information carriers are available for the public to choose and use. The way people obtain information has changed from traditional media such as radio and television to new self-media platforms such as Post Bar and TikTok. Because of advancements in Internet technology, college students now have access to an increasing number of venues that allow them to communicate their thoughts and feelings and elaborate on their positions. Relying on the self-media platform, netizens can quickly obtain information and pay attention to the development of the whole event. They can have a lively discussions on the information and news events published on the Internet, and express their own opinions. At present, the self-media platform has occupied an irreplaceable leading position in the dissemination of social information. Currently, the data is showing explosive growth. We-media has become a vital way for current internet users to watch the world, comprehend society, investigate themselves, and express their needs. These features include convenience, autonomy, and equality. As the main force in the huge group of netizens, college students in adolescence not only have the characteristics of impulsiveness and sensitivity, but also are interested in new things and are easy to accept. However, due to incomplete ideology and single learning experience, college students are easy to be used. College administrators can only genuinely join the lives of students and design policies that are targeted at their needs if they have a current awareness of the ideological dynamics of college students as well as a comprehensive understanding of the concerns that college students have. It is of the utmost importance to gain an accurate understanding of the ideological dynamics of college students and to increase the monitoring and management of the concerns of college students (*Velsor-Friedrich & Hogan, 2021*; *Junaid et al., 2021*).

Sentiment analysis of college students (*Beasley, Friedman & Rosen, 2021*; *Srinivas & Rajendran, 2019*; *Kornbluh et al., 2021*) is very important for schools to manage students, and the core of sentiment analysis is accurate emotion recognition. At present, the research related to emotion recognition has been relatively mature. In the aspect of sentiment classification based on sentiment dictionary, the literature (*Yang et al., 2013*) constructed a Japanese-based sentiment dictionary. Literature (*Hassan, Usman & Saba, 2016a*; *Hassan, Usman & Saba, 2016b*; *Hassan, Usman & Saba, 2016c*) used different machine learning algorithms (*Lee, 2020*; *Fornwalt & Pfeifer, 2021*; *Paixao et al., 2022*) to integrate with emotion dictionaries to improve the accuracy of emotion recognition. Some scholars use machine learning algorithms alone for sentiment classification, which is faster. Literature (*Qiang, Zhang & Law, 2009*) combined support vector machine and Bayesian machine learning models for sentiment analysis of text information. Literature (*Hua & Dong, 2018*) uses a naive Bayesian classification algorithm to perform sentiment analysis on microblog data. A method for cross-domain text sentiment analysis is proposed in the aforementioned piece of literature (*Boiy & Moens, 2009*), and it takes into account the unique properties of texts from other domains. The issues of text noise characteristics and the size of the training set are both efficiently resolved with this strategy. It is also very

common practice to apply deep learning algorithms when conducting sentiment analysis. An N-gram model is proposed in the aforementioned piece of scholarly writing (*Sharma & Srivastava, 2021*), and it is classified by stacking three-layer neural network models. This significantly reduces the dimension of word vectors. The problem of sentiment classification is solved using the long-short-term memory model (LSTM) in the cited piece of literature (*Habernal, Ptacek & Steinberger, 2014*). LSTM has been enhanced based on RNN, and it is now able to successfully obtain contextual information. A combined model of neural networks is proposed in the aforementioned piece of literature (*Tai, Socher & Manning, 2015*). When training word embedding representations using this model, the contextual information features learned by LSTM and the features gained by CNN are mixed in order to provide features that are more appropriate for the text. This approach is utilized in a variety of different categories of feelings. An innovative concept for further study has also been presented in addition to the substantial progress that has been made in the work.

There are several problems with the above text sentiment analysis methods. First, although the effect of sentiment analysis is relatively accurate, the model used is complex and the time complexity is high. Second, although the model is simple and easy to implement, the effect of sentiment analysis is general. Third, the object of sentiment analysis is not aimed at specific groups and has no practical reference value. Based on the above situation, this article intends to use an improved CNN model for text emotion recognition, so as to obtain a text emotion analysis system for college students. The following is a list of the primary contributions that this article makes: (1) use the text separable convolution algorithm to perform hierarchical convolution on text features to achieve refined extraction of word vector information and context information. Doing so avoids semantic confusion and reduces the complexity of convolutional networks. (2) The text separable convolution algorithm is applied to text-based sentiment analysis, and an improved CNN model is further proposed. (3) The experimental section of the study compares the model that was used to evaluate text with other models in order to determine how effective the model was at determining the sentiment of the text. According to the findings of the experiments, the model that was utilized has superior performance in the text-based sentiment analysis task.

## MATERIALS & METHODS

### Typical text sentiment analysis methods

The fundamental component of text sentiment analysis is the classification of sentiments according to the information that is expressed by text data. Standard approaches to sentiment analysis can be broken down into three distinct categories: those that are founded on sentiment dictionaries, those that are founded on machine learning, and those that are founded on deep learning. The use of sentiment dictionary method for sentiment classification is the most primitive technical means in text sentiment classification. The advantage of this method is its flexibility, because we can build different sentiment dictionaries according to different scenarios and different needs. However, as a result of

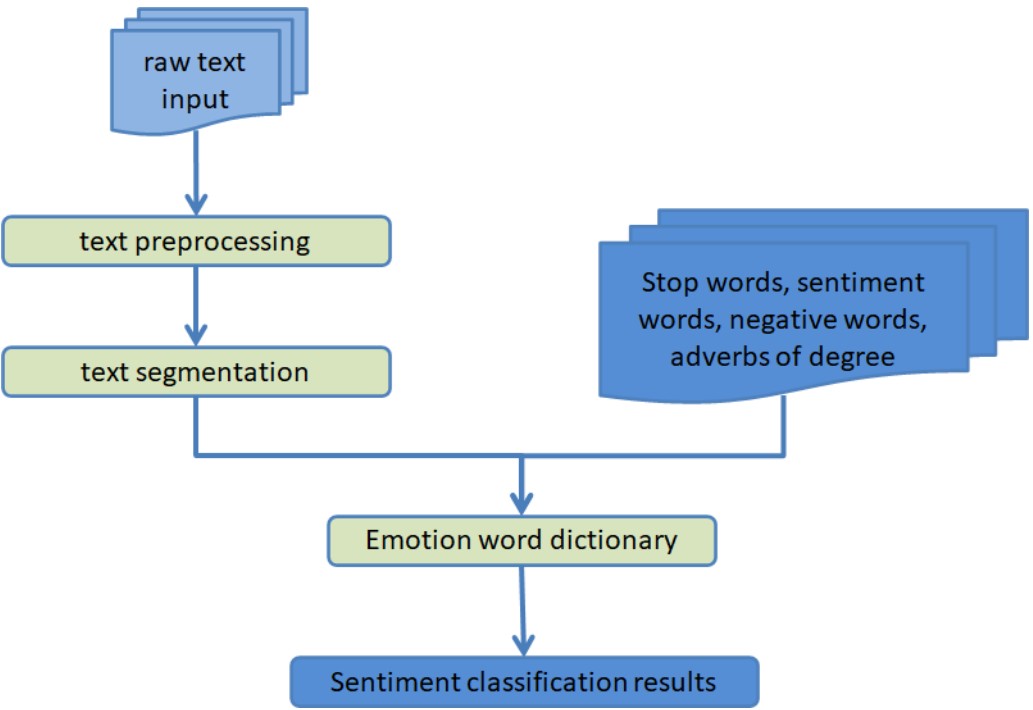

**Figure 1** Schematic diagram of text sentiment analysis based on sentiment dictionary.

advances in technology and shifts in social mores, the text is becoming progressively less structured, and the emotional underpinnings of the majority of texts are no longer readily apparent. The use of a sentiment dictionary for the purpose of sentiment classification has not produced the effect that users had anticipated; hence, at this time, the method of sentiment classification that is based on machine learning was developed and is now extensively utilized. The text sentiment analysis shown in Fig. 1 is presented in the form of a schematic diagram based on sentiment dictionary.

When compared to the method that is based on a sentiment dictionary, the method that is based on machine learning has a significant improvement in the effect of text sentiment classification; however, both methods show their respective advantages in a variety of tasks that involve sentiment classification. However, each of these approaches do have some potential downsides. The update speed of the method of a sentiment dictionary is slow and cannot keep up with the times, which means that the accuracy of sentiment classification is closely related to the size of a sentiment dictionary and the accuracy of manual annotation. In addition, the size of a sentiment dictionary directly influences the accuracy of manual annotation. There is a significant amount of human processing effort involved in the feature engineering of sentiment categorization systems that are based on machine learning. When there is an excessive volume of data, the efficiency will be very poor or it may even be impossible to achieve. Despite the fact that the machine learning model is straightforward, it has a limited capacity for generalization and a weak

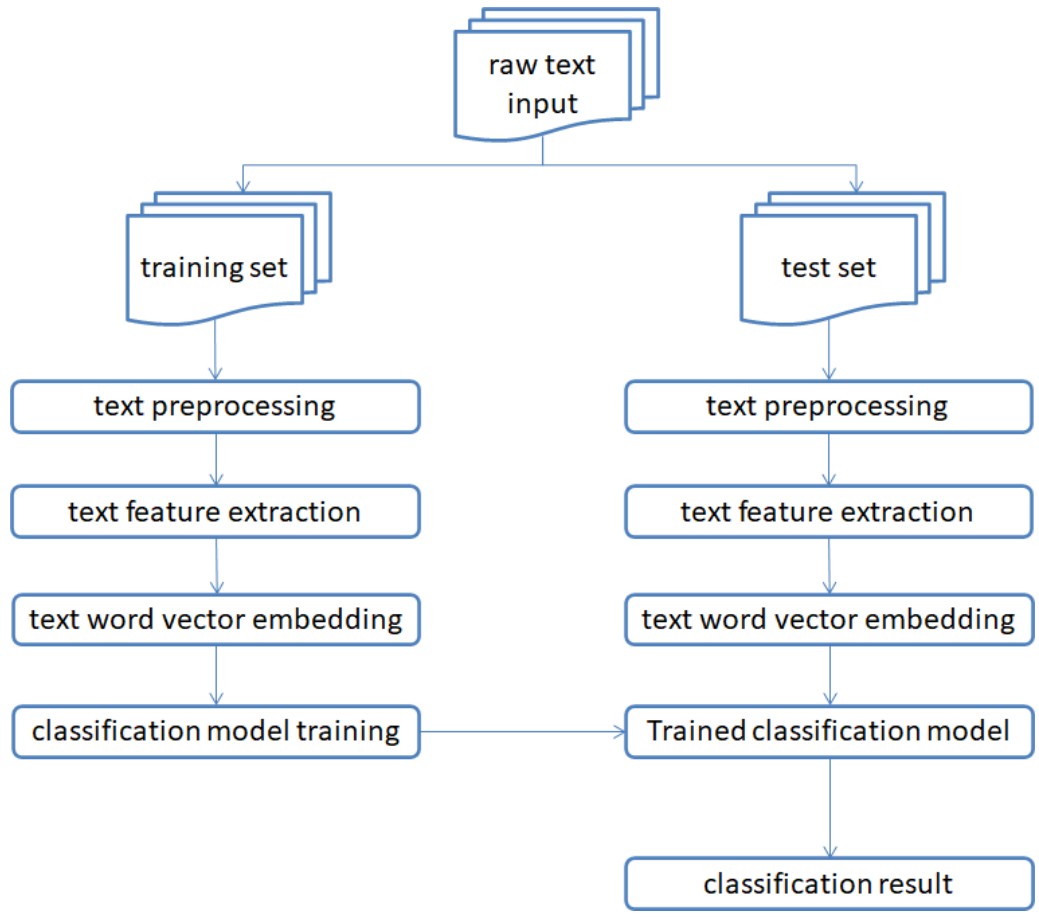

**Figure 2  A diagram of how machine learning is used to analyze the mood of text.**

classification effect. The text sentiment analysis based on machine learning is shown here by a schematic, which can be found in Fig. 2.

Deep learning algorithms outperform sentiment dictionaries and machine learning algorithms in classification performance. However, deep learning algorithms also have their inherent problems, such as time-consuming algorithm operation, high algorithm complexity, too many parameters to be adjusted, and algorithm sensitivity to parameters. Scholars have also successively optimized and improved deep learning algorithms from the above levels to further improve their sentiment analysis effects on texts. Figure 3 presents the results of a sentiment analysis of text based on deep learning.

## Convolutional neural network model

The CNN is an example of a feedforward neural network, which is a type of neural network that is highly representative of deep learning. Compared with other neural networks, the convolution and pooling structures of CNN require relatively few parameters to debug, and have excellent performance in sentence matching recommendation systems, semantic processing, text classification and other fields. In 2014, *Kim (2014)* proposed the Text-CNN

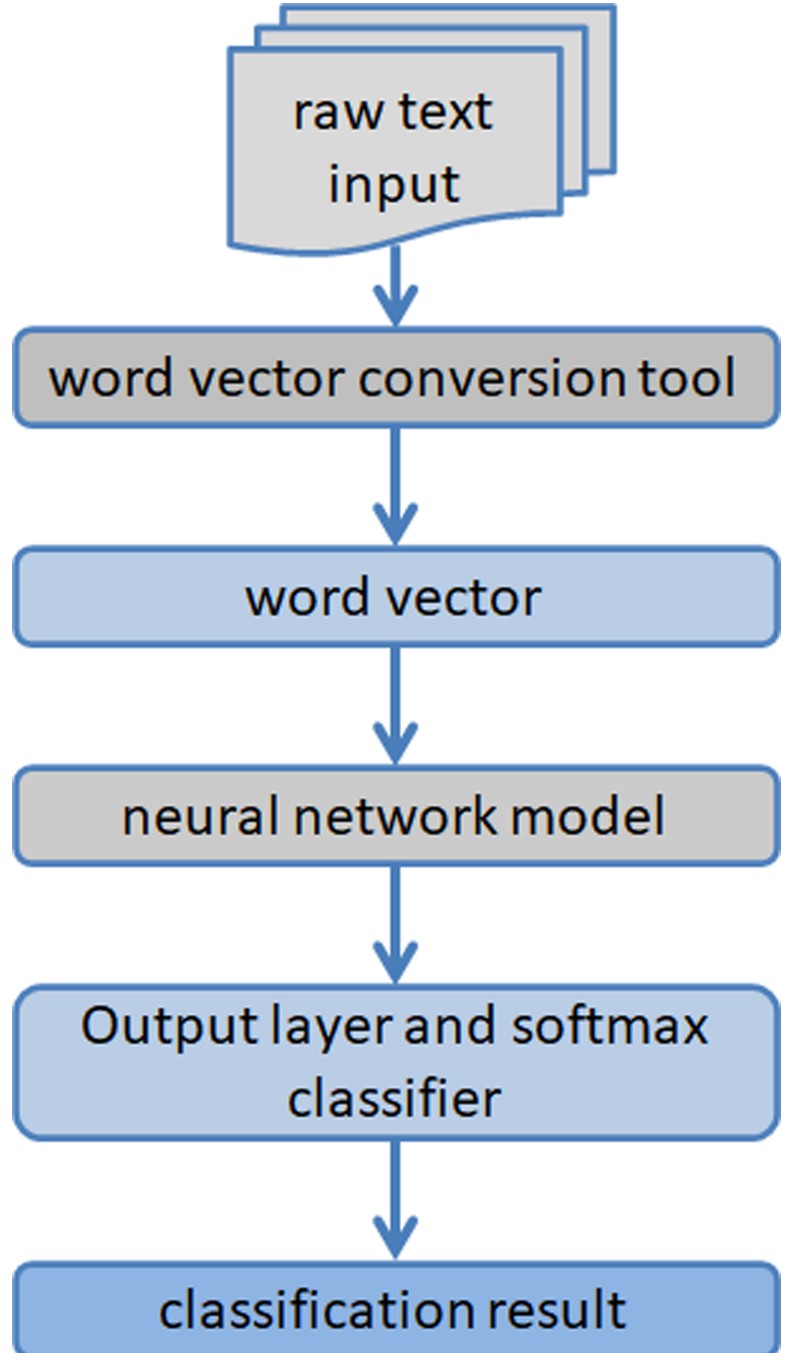

**Figure 3** A diagram of how deep learning is used to analyze the mood of text.

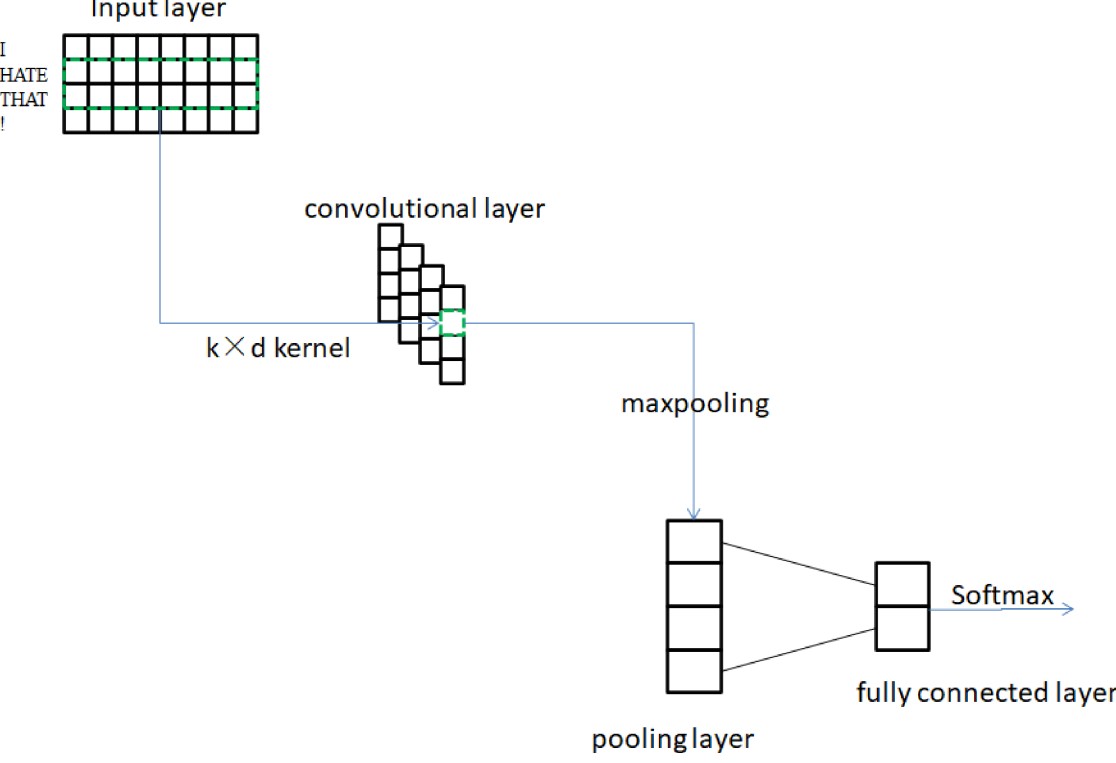

**Figure 4  Text-CNN structure.**

model, which uses multiple convolution kernels to extract features similar to N-gram for text, and achieved good results in the field of short text classification. The overall network architecture of the model is shown in Fig. 4.

The Text-CNN model is made up of four distinct components, which are the input layer, the convolution layer, the pooling layer, and the fully connected layer. The matrix that corresponds to the document serves as the input for the model at the model input layer. Generally, the word is first mapped into a word vector with a fixed length of d through the Word2Vec model, and then the document is truncated or supplemented into a text with a length of l. In this way, the entire document can be represented in the form of a matrix of size l * d. The contextual N-gram features of sentences are then captured by convolutional layers. The difference from the commonly used image convolution kernels is that the Text-CNN model transforms the traditional k × k convolution kernels into k × d convolution kernels suitable for text classification, where k is the convolution kernel window size. This can ensure that the word vector information will not be truncated during the convolution process and maintain the integrity of the semantics. The feature map that is produced by the convolutional layer is then sent to the pooling layer, where the downsampling operation is used to select features and filter information in order to accomplish dimensionality reduction, the removal of redundant information, and the simplification of network complexity.

## Text sentiment analysis model based on improved depthwise separable convolution neural networks

### *Improved depthwise separable convolution neural networks*

A word vector is a comprehensive reflection of the meaning of a word. Unlabeled data training is destined to be a semantic representation that combines polysemy and all contextual information. So even the simplest linear model, such as Fast-Text. The model can also achieve good classification results. In view of the problem that the existing deep learning model cannot make full use of the word vector features and the model complexity is too high, this research divides the traditional convolutional layer into two layers, one layer performs word embedding convolution, extracts word vector features, and the other layer perform regional convolution to extract contextual features.

The depthwise separable convolution algorithm first appeared in the Mobile Net (*Kazerouni, Dooly & Toal, 2021*) model in the field of image classification. In *Qin et al. (2019)*, a decomposable convolutional network is used to realize the lightweight conversion of the traditional SSD network, which greatly reduces the complexity of the network. Figure 5 presents the depthwise decomposable convolution structure in the Mobile Net model. The model employs depthwise convolution filters and $1 \times 1$ convolution filters instead of traditional image standard convolution filters. But the image convolutional network structure is not suitable for text. Since for an image, the correlation between any pixel and surrounding pixels is probabilistically equal, it is feasible to use a convolution kernel of size $C_w \times C_w$. However, for text, the word vector of a word and the context in which the word is located are completely different considerations, and the two cannot be confused. Therefore, it is inappropriate to use a convolution kernel of size $C_w \times C_w$ or $C_w \times d$, where $C_w$ represents the size of the convolution kernel, and $c$ represents the dimension of the word vector.

Aiming at the structural features of text and the idea of depthwise separable convolution, this study proposes a separable convolution algorithm suitable for text. First, the text separable convolutional network retains the advantage of low complexity of the image separable convolutional network. Second, the model realizes the hierarchical extraction of text features, which ensures the integrity of the word vector information during the convolution operation. It can make full use of text features and reduce the semantic ambiguity caused by polysemy and other phenomena. Figure 6 shows how the standard text convolution filters are decomposed into word embedding convolution filters and region convolution filters.

First, after text preprocessing and vectorization operations, the size of the sentence is $m \times c \times 1$. The standard text convolution kernel size is $C_w \times c \times 1 \times S$, where $m$ is the text length, $c$ is the word vector dimension, and $C_w$ is the window size of the convolution kernel, $S$ is the number of output channels. Therefore, the computational cost of text standard convolution is

$$m \times c \times 1 \times C_w \times c \times 1 \times S. \tag{1}$$

The text separable convolution algorithm emphasizes that the connection between the dimensions of the word vector is different from the contextual connection between words.

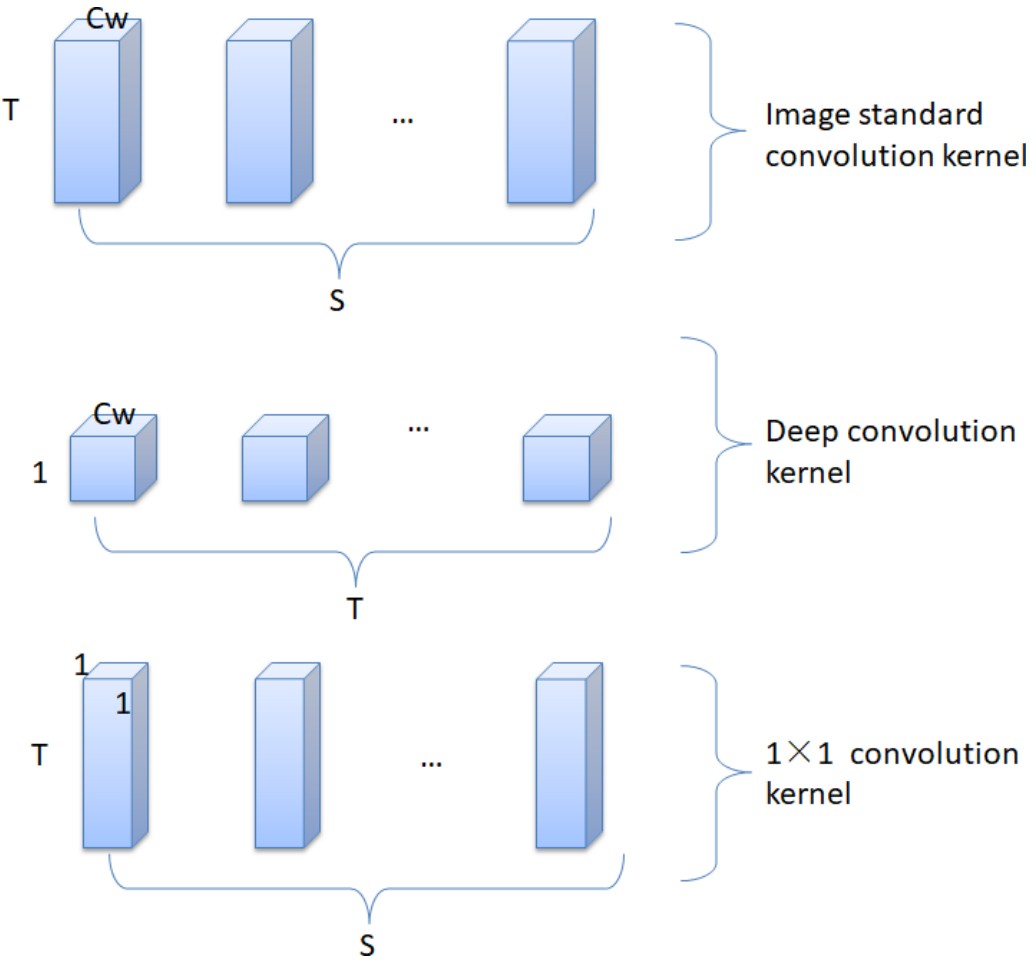

**Figure 5** **Image decomposable convolution structure.**

Therefore, first use word embedding convolution to break the interaction between regional convolutions, so that the convolution kernel only extracts a single word vector feature. In this way, the model only pays attention to the word itself in the convolution operation, so as to obtain as much word vector information as possible, so that the model can learn to distinguish the phenomenon of polysemy. Second, regional convolution is applied to obtain regional features of each word to analyze the local content of its context. The word embedding convolutional layer uses a convolution kernel with a size of $1 \times c \times 1 \times T$, and the calculation amount of the convolution process is:

$$m \times c \times 1 \times 1 \times c \times 1 \times T. \tag{2}$$

After the first iteration of the convolutional algorithm, the size of the feature map that is obtained is $1 \times 1 \times T$. In the second step, the size of the regional convolution kernel is calculated as follows: $C_w \times 1 \times T \times S$, where T is the number of word embedding convolution kernels and S is the number of regional convolution kernels. In other words, the size of the regional convolution kernel is determined by the product of these two numbers.

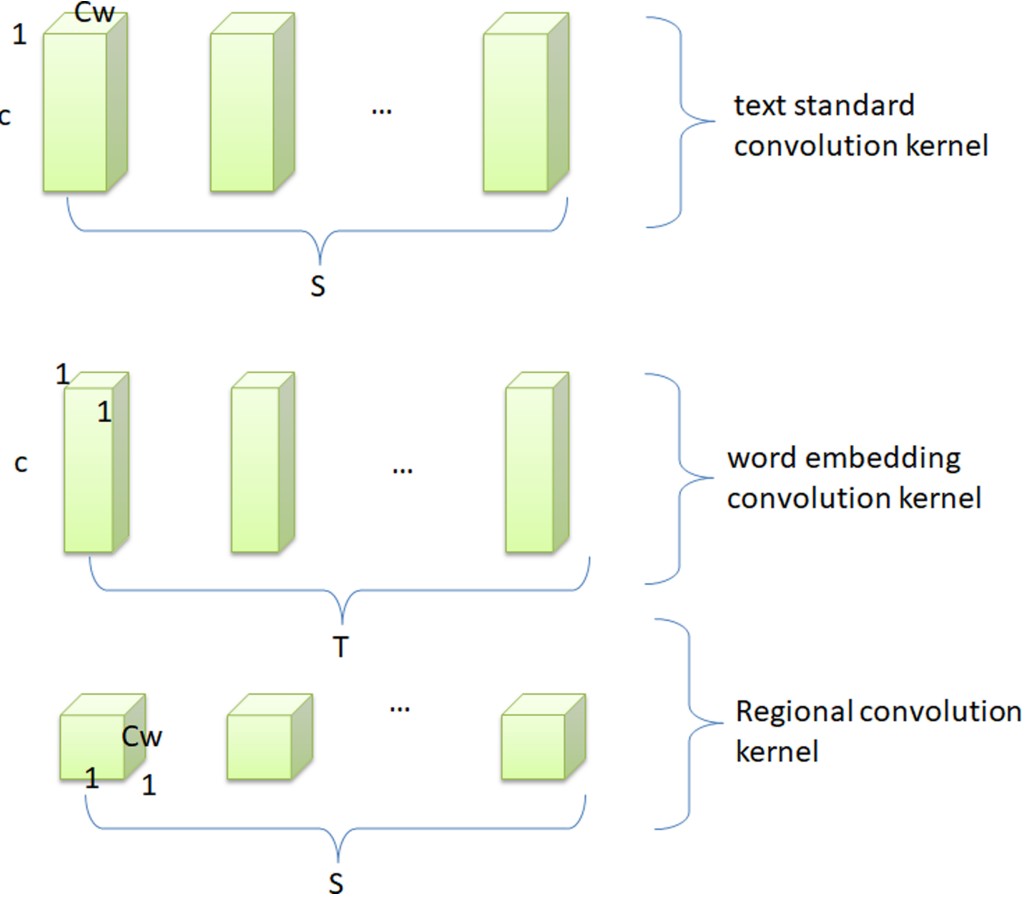

**Figure 6** Text decomposable convolutional structure.

Applying this to the feature map above, the computational cost of this convolution process is:

$$m \times 1 \times 1 \times T \times C_w \times 1 \times T \times S. \tag{3}$$

Therefore, the total computational cost of the text decomposable convolutional network is:

$$m \times c \times c \times T + m \times T \times C_w \times T \times S. \tag{4}$$

By simplifying Eq. (1), the complexity of the traditional text convolutional network becomes:

$$m \times c \times C_w \times c \times S. \tag{5}$$

Therefore, by expressing the traditional text convolution as a two-step convolution process of word embedding convolution and region convolution, the reduction in computation is as follows:

$$\frac{m \times c \times C_w \times c \times S}{m \times c \times c \times T + m \times T \times C_w \times T \times S} = \frac{T}{S \times C_w} + \frac{T}{c \times c}. \tag{6}$$

**Peer**J Computer Science

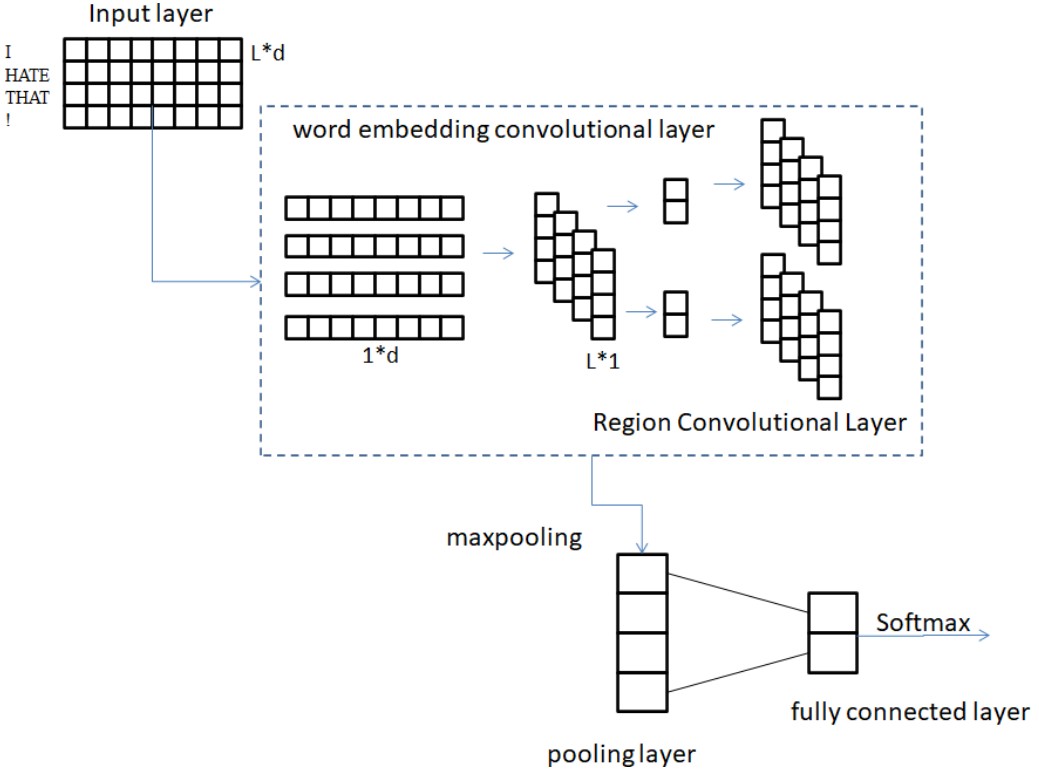

**Figure 7** Structure of text sentiment analysis model.

Taking $S = T$, and $S$ is the same order of magnitude as $c$, and $C_w$ is much smaller than $c$, Eq. (6) is simplified as follows:

$$\frac{T}{S \times C_w} + \frac{T}{c \times c} \approx \frac{1}{C_w}. \tag{7}$$

Equation (7) shows that the complexity of the text separable convolution algorithm is $\frac{1}{C_w}$ of the traditional convolution algorithm. This shows that the practical application value of the separable convolution network in the field of text analysis and processing is much higher than that of the traditional convolution algorithm from the perspective of mathematical analysis. Therefore, from the perspective of model complexity, the text sentiment analysis model based on separable convolutional network is more superior than other deep learning models.

## Text sentiment analysis model

A model for analyzing the sentiment of text is built by combining the detachable convolutional network described in Section 3.1 with Softmax. Figure 7 provides a visual representation of the structure of the text sentiment analysis model.

Five layers make up the entire model: the input layer, a word embedding convolutional layer, a region convolutional layer, a pooling layer, and a fully connected layer. The conventional CNN methodology is used for the processing of the first input layer here. The preprocessed sentence is mapped into a vector matrix through the Word2Vec model,

which is convenient for subsequent convolution operations. Let $x_i \in R^c$ denote the word in the sentence with the $i$ th vector of dimension $c$. $x \in R^{m \times c}$ denote the input sentence with sentence length $m$. Therefore, the training text of length $m$ can be expressed as:

$$x = [x_1, x_2, \ldots, x_m]. \tag{8}$$

The purpose of the second layer of word embedding convolutional layers is to handle word sense and contextual meaning separately. Doing so enables the convolution kernel to focus on analyzing the semantic content of words, thereby extracting fine-grained textual features. This part of the word embedding convolution operation weight $Z1 \in R^{1 \times c}$. A convolution kernel is applied to a single word vector to generate new features that contain only information about the single word. As an illustration, a word feature $f1$ is produced when a word $x_i$ is used:

$$f1_i = h(Z1 \times x_i + g1) \tag{9}$$

where $g1 \in R$ is the bias term, which is mainly used for the word embedding convolution kernel. $h$ refers to nonlinear functions such as tangent, sigmoid, etc., because RELU can avoid gradient explosion and gradient disappearance within a certain range. In the model, RELU is used as the activation function $h$. Equation (9) is applied to all word embedding convolution kernels to generate feature map $f1$:

$$f1 = [f1_1, f1_2, \ldots, f1_m]. \tag{10}$$

$f1 \in R^m$, where $f1$ represents the word vector information feature of the text after word embedding volume. The third layer is the regional convolution layer, whose purpose is to obtain the contextual features of words based on the word information of the previous layer. The filter $Z2 \in R^{C_w \times 1}$ of this partial region convolution operation is applied to the c1 feature map to capture the contextual features of each word. For example, feature $f2_i$ is generated from window $[f1_i, \ldots, f1_{i+C_w-1}]$ by Eq. (11):

$$f2_i = h(Z2 \times [f1_{i:i+W_c-1}] + g2). \tag{11}$$

$g2$ is a bias term for the $f1$ feature map. The activation function $h$ adopts the RELU function. The filter is applied to each possible window of the word embedding feature, and the resulting feature map is as follows

$$f2 = [f2_1, f2_2, \ldots, f2_{m-W_c+1}]. \tag{12}$$

Where $f2 \in R^{m-C_w+1}$, which represents all the contextual information features of the text after regional convolution.

The fourth layer of pooling layer, its purpose is to further extract f2 features and realize feature dimensionality reduction. Improve model generalization ability by reducing model complexity. The most common pooling operations are average pooling and max pooling. In this article, the maximum pooling is selected, that is, the maximum value in the feature map is selected as the final value after pooling in this area, because the maximum pooling can reduce the estimated mean shift caused by the error of the convolutional layer parameters.

Through the pooling layer, the most important context information is selected to represent the entire text feature, and several one-dimensional vectors are obtained. $\widetilde{f}2$ isused to represent the max pooling feature:

$$\widetilde{f}2 = \max\{c2\}. \tag{13}$$

Due to the existence of the number of regional convolution kernels, $\widetilde{f}2 \in R^N.\widetilde{f}2$ characterizes the finally obtained comprehensive text features. The last layer is the fully connected layer. The $\widetilde{f}2$ feature applies dropout to the fully connected layer, and the layer weight vector $Z3$ is constrained by L2 regularization. Dropout can make the activation probability of some neurons fixed on the $e$ value, and the $e$ value ranges from 0 to 1. Therefore, the model does not depend too much on some local features in the forward transmission process, which makes the model more robust and generalizable. Added L2 regularization to prevent overfitting more effectively. That is, Eq. (15) can be used instead of Eq. (14) to represent the output unit $y$ of the forward transmission.

$$y = Z3 \times \widetilde{f}2 + g3 \tag{14}$$
$$y = Z3 \times (\widetilde{f}2 \cdot v) + g3 \tag{15}$$

where $g3$ is also a bias term. $\cdot$ isthe element-wise multiplication operator. $v \in R^S$ isthe masking vector of the Bernoulli random variable. A probability of 1 indicates dropout, realizing that gradients are propagated only through unmasked units. Finally, the output unit $y$ gets the final classification label through the Softmax classifier.

## EXPERIMENTAL ANALYSIS AND DISCUSSION

### Experimental setup

This article plans to grasp the emotional state of college students through text emotion recognition, so as to better assist colleges and universities to manage college students' study and life in school. As a result, the text sentiment data of college students was compiled by the authors of this research in order to validate the usefulness of the model that was applied. On the other hand, in order to conduct a more objective analysis of the performance of the model that was utilized, the experimental analysis on public datasets that this article presents is carried out. Whether it be an experiment using a publicly available dataset or one with a dataset that the author has created themselves, the experimental environment described in this study adheres to the environment provided in Table 1:

### Public dataset experiments

The public dataset used in this article is Chnsenticorp. The number of sentiment classifications for this dataset is 2, positive and negative. The sentiment label of positive text is 1, and the sentiment label of negative text is 0. Some experimental data samples are shown in Table 2.

A total of 60% of the dataset is used as the training dataset and 40% as the test dataset. The evaluation indicators use Accuracy, Precision, and Recall. Several classic deep learning

**Table 1  Description of experimental environment.**

| Experimental environment | Details |
|---|---|
| Software | Programming language: Python3.7 |
| | Development platform: Anaconda3 |
| | Third-party libraries: jieba, Numpy, Pandas |
| Hardware | Operating System: Windows 10 Professional |
| | Processor: Intel(R) Core(TM) i3-3120M CPU @2.50 GHZ |
| | Memory: 8G |
| | Hard disk: 1T |

**Table 2  Example of experimental data.**

| Text details | Sentiment labels |
|---|---|
| Very nice hotel, I have stayed many times. | 1 |
| The hotel facilities are very old, the bathroom is really dirty, and there is no elevator. | 0 |
| The room is too small, other facilities are average. | 0 |

**Table 3  Experimental results on public datasets.**

| Index\Model | CNN | RNN | LSTM | BiLSTM | Text-CNN | Proposed |
|---|---|---|---|---|---|---|
| Accuracy | 0.8576 | 0.8632 | 0.8465 | 0.8617 | 0.8963 | 0.9136 |
| Precision | 0.8375 | 0.8376 | 0.8326 | 0.8485 | 0.8694 | 0.8962 |
| Recall | 0.8023 | 0.8135 | 0.8022 | 0.8112 | 0.8337 | 0.8688 |

models are selected for the comparison model, namely CNN (*Enamoto, Li & Rocha, 2021*), recurrent neural network (RNN (*Kim, Jo & Lee, 2019*), LSTM (*Schmitz et al., 2021*), BiLSTM (*Noh & Cho, 2021*), Text-CNN (*Zhang et al., 2021*)). Table 3 displays the results of the experiments conducted on each model using the publicly available dataset:

Following are some of the findings that can be drawn from an examination of the data shown in Table 3: (1) for the Accuracy indicator, the results obtained by each model exceeded 0.85. The experimental data obtained by several classical algorithms such as CNN, RNN, LSTM, and BiLSTM were similar, and the BiLSTM model obtained the best results. BiLSTM adds a gate mechanism and memory unit on the basis of RNN, which effectively prevents gradient explosion and gradient disappearance. At the same time, it better captures long-distance dependencies, and BiLSTM can capture bidirectional semantic dependencies. This does, in fact, result in an improvement in the overall performance of the model. Text-CNN has implemented specific enhancements aimed towards the text data. The results of the experiments make it abundantly evident that the suggested model has vastly improved the classification results of text data; specifically, it has improved by 1.93% on the basis of Text-CNN. This can be seen clearly from the results of the experiments. (2) The Precision index values obtained by each model are lower than the Accuracy value. The reduction of RNN is significantly lower than that of CNN, which is why this article chooses CNN as the basic model. For the Text-CNN model that performs well on the Accuracy

**Table 4  Experimental results on self-made datasets.**

| Index\Model | CNN | RNN | LSTM | BiLSTM | Text-CNN | Proposed |
|---|---|---|---|---|---|---|
| Accuracy | 0.9127 | 0.9032 | 0.9195 | 0.9338 | 0.9363 | 0.9576 |
| Precision | 0.8994 | 0.8839 | 0.9011 | 0.9296 | 0.9297 | 0.9501 |
| Recall | 0.8835 | 0.8787 | 0.8940 | 0.8961 | 0.9008 | 0.9364 |

indicator, its Precision indicator value is obviously not ideal. The model used in this article is improved by 3.08% on the basis of Text-CNN. (3) For the Recall indicator, the proposed model is improved by 8.29%, 6.8%, 8.3%, 7.1%, and 4.21%, respectively, compared with CNN, RNN, LSTM, BiLSTM, and Text-CNN. Good performance shows that the model has good robustness.

### Self-made dataset experiment

This study crawled the data in the on-campus post bar of a university in Jiangsu Province, and randomly selected 1,000 posts from the data. A total of four people categorize posts sentimentally into two categories, positive and negative. This post data is kept only if four people give the same category tag, otherwise, the post will be discarded. Finally, 16,268 posts were retained as experimental data. There are 12,000 posts that are chosen at random to be used as training data, while the remaining posts are used as test data. The comparative model agrees with the model that was utilized in the experimental data that is freely available to the public. The results of the experiments performed on the data collected by oneself are presented in Table 4 as follows:

In general, the experimental findings on the self-made data set are superior to the experimental results on the public data set. This is due to the fact that the emotional tendency of the text data selected when making the data is pretty clear. All of the models have a recognition accuracy that is more than 0.9. This demonstrates that deep learning models are capable of achieving very good outcomes in the process of analyzing the sentiment of text. RNN has the worst performance when it comes to the recognition performance of the self-made dataset. The experimental findings achieved by the BiLSTM model are comparable to those obtained by Text-CNN. The suggested model exhibits an increased accuracy of 2.55% and 2.27% when compared to the BiLSTM and Text-CNN models, as well as an improvement in precision of 2.21% and 2.19%, and an increase in recall of 4.5% and 3.95% respectively. All three of the assessment indexes point to the fact that the suggested model has the most successful influence on emotion recognition in written text. This shows that the proposed model performs hierarchical convolution on text features, and can indeed achieve refined extraction of word vector information and context information. Doing so avoids semantic confusion and reduces the complexity of convolutional networks. The improvement of the model can solve the text sentiment analysis task well, which further verifies the effectiveness of this work.

## CONCLUSION

For a long time, understanding and mastering the thoughts and emotions of college students in a timely manner is content that college administrators are very concerned about. It is

a popular trend to understand the concerns of college students by mining social data of college students. With the explosive growth of massive data, self-media platforms have gradually become the main place for college students to disseminate information around them in a timely manner, and to publicly publish their life experiences and speeches in colleges and universities. Therefore, it is of great significance to conduct sentiment analysis on the concerns of college students. A text sentiment analysis method is proposed in this article in order to achieve a deeper understanding of the psychological and emotional dynamics that are present among college students. The classification of the sentimental tendencies of text data is accomplished by the utilization of an enhanced convolutional neural network by the system, which in turn allows for an analysis of the emotional dynamics of the individuals who published the text. The most important points discussed in this article can be summed up as follows: In order to accomplish the refined extraction of word vector information and context information, the text features are first convoluted hierarchically by using the text separable convolution technique. When this is done, semantic confusion can be avoided, and the complexity of convolutional networks can be reduced. Second, an enhanced convolutional neural network model is proposed, and the text separable convolution technique is used to the process of analyzing the sentiment of text. When compared to other models, the performance of this model in the text sentiment analysis task is significantly superior. However, there are several issues that still need to be improved upon in this article in order to make it more optimal. First and foremost, the number of sentiment classifications that may be used for the work of conducting sentiment analysis with college students can also be refined. There is the potential for the development of neutral emotion categories in addition to the positive and negative emotion categories. When data is collected, it should also be more normalized as much as possible. The data processing has a significant influence on the performance of the model due to the rather colloquial nature of the speech in the post bar as well as the erratic structure of the data. Second, the model should be simplified as much as possible so as to lessen the complexity of its runtime. Third, the model's recognition performance should be improved so that it can be used to actual production as much as possible. This will allow the model to have the most impact.

### Funding
This study is sponsored by the 2022 Jiangsu University Philosophy and Social Science Research General Project "Research on the Realistic Dilemma and Technical Appeals of University Precision Funding from the Perspective of Big Data" (Fund No.: 2022SJYB1050). The funders had no role in study design, data collection and analysis, decision to publish, or preparation of the manuscript.

### Grant Disclosures
The following grant information was disclosed by the authors:

Jiangsu University Philosophy and Social Science Research General Project "Research on the Realistic Dilemma and Technical Appeals of University Precision Funding from the Perspective of Big Data": 2022SJYB1050.

## Competing Interests

The authors declare there are no competing interests.

## Author Contributions

- Xiaoyu Kong conceived and designed the experiments, performed the experiments, analyzed the data, performed the computation work, prepared figures and/or tables, authored or reviewed drafts of the article, and approved the final draft.
- Ke Zhang performed the experiments, analyzed the data, performed the computation work, prepared figures and/or tables, authored or reviewed drafts of the article, and approved the final draft.

## Data Availability

The data is available at GitHub: https://github.com/SophonPlus/ChineseNlpCorpus/tree/master/datasets/ChnSentiCorp_htl_all.

## Supplemental Information

Supplemental information for this article can be found online at http://dx.doi.org/10.7717/peerj-cs.1236#supplemental-information.

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
