# Peer review of "A novel text sentiment analysis system using improved depthwise separable convolution neural networks"

_PeerJ Computer Science, doi:10.7717/peerj-cs.1236_

## Round 0.1 · original submission · Major Revisions

This paper needs a major revision. The authors are asked to revise their work according to the obtained comments.

·

Basic reporting

An improved convolutional neural network is proposed by the author as a means of reading the author's intended feelings from the text. Positive results were observed when emotion analysis was applied to text, as shown by the experiment. The beneficial societal significance of this study is mostly attributable to the emotional analysis of college students. Nonetheless, the author's writing skills may use some work. The article also contains some material that isn't very detailed and, in some cases, completely omits information that is crucial to understanding the subject at hand. For the most part, I'm optimistic that these problems will be fixed in the new version.
(1) The author provides a suggestion for a totally new approach of studying feelings based on language, which they call the linguistic analysis of feelings. As the primary learning method, an improved version of the core algorithm for deep learning is utilized. Why does the author propose a novel approach to the investigation of emotions, and what are the motivations behind this proposal? The study of feelings can currently be approached from a number of different angles, each of which is predicated on the concept of deep learning.
(2) When it comes to the quality of the language, there is certainly potential for advancement. It is commonly believed that certain phrases, such as the ones that are listed below, originated in China. " During the training process for word embedding representations using this model, the contextual information features obtained by LSTM and the features gained by CNN are combined in order to generate features that are better suited for the text. These features are generated by blending together the features gained by LSTM and CNN. This tactic is utilized in a wide variety of different types of expressions all over the board. In addition to the significant headway that has been made in the study, a fresh concept for doing additional research has also been proposed. In addition to the progress that has already been accomplished in the effort, this is a new development."
(3) At the moment, the system for assessing emotions takes use of a wide variety of resources, including video, facial expression, text, and electroencephalogram (EEG), to name a few. Why did the author decide to undertake an emotional analysis on this specific work of literature as opposed to another one?
(4) In the section of the report that is devoted to the introduction, research on affective analysis that is based on deep learning is given. It is important, when discussing the study that is relevant to the topic at hand, to explain the fundamental differences between the emotion analysis system based on convolutional neural networks that was proposed by the author and these studies. This can be done when discussing the study that is relevant to the topic at hand.
(5) The author has devised a method that is superior to the traditional convolutional neural network in terms of its overall effectiveness. What is the main concept behind using the upgraded network when it comes to the analysis of the emotions included within text? It has been brought to our attention that the graphic description may benefit from some more color.
(6) Sixty percent of the dataset is used as the training set, and forty percent of the dataset is used as the test set, according to the author's description of the use of the dataset, which is not totally apparent. How to determine which sets should be used for testing and which should be used for training is the question that needs to be solved. Is it simply a matter of chance, or could there be another factor at play here?
(7) The following is the computation that needs to be done in order to determine the outcome of the experiment: Is this the result of a single experiment or a collection of findings from a number of other tests carried out separately? The piece does not offer any kind of clarification on this matter at all.

Experimental design

- The author has devised a method that is superior to the traditional convolutional neural network in terms of its overall effectiveness. What is the main concept behind using the upgraded network when it comes to the analysis of the emotions included within text? It has been brought to our attention that the graphic description may benefit from some more color.
- Sixty percent of the dataset is used as the training set, and forty percent of the dataset is used as the test set, according to the author's description of the use of the dataset, which is not totally apparent. How to determine which sets should be used for testing and which should be used for training is the question that needs to be solved. Is it simply a matter of chance, or could there be another factor at play here?
- The following is the computation that needs to be done in order to determine the outcome of the experiment: Is this the result of a single experiment or a collection of findings from a number of other tests carried out separately? The piece does not offer any kind of clarification on this matter at all.

Validity of the findings

An improved convolutional neural network is proposed by the author as a means of reading the author's intended feelings from the text. Positive results were observed when emotion analysis was applied to text, as shown by the experiment. The beneficial societal significance of this study is mostly attributable to the emotional analysis of college students. Nonetheless, the author's writing skills may use some work. The article also contains some material that isn't very detailed and, in some cases, completely omits information that is crucial to understanding the subject at hand.

Additional comments

For the most part, I'm optimistic that these problems will be fixed in the new version.
(1) The author provides a suggestion for a totally new approach of studying feelings based on language, which they call the linguistic analysis of feelings. As the primary learning method, an improved version of the core algorithm for deep learning is utilized. Why does the author propose a novel approach to the investigation of emotions, and what are the motivations behind this proposal? The study of feelings can currently be approached from a number of different angles, each of which is predicated on the concept of deep learning.
(2) When it comes to the quality of the language, there is certainly potential for advancement. It is commonly believed that certain phrases, such as the ones that are listed below, originated in China. " During the training process for word embedding representations using this model, the contextual information features obtained by LSTM and the features gained by CNN are combined in order to generate features that are better suited for the text. These features are generated by blending together the features gained by LSTM and CNN. This tactic is utilized in a wide variety of different types of expressions all over the board. In addition to the significant headway that has been made in the study, a fresh concept for doing additional research has also been proposed. In addition to the progress that has already been accomplished in the effort, this is a new development."
(3) At the moment, the system for assessing emotions takes use of a wide variety of resources, including video, facial expression, text, and electroencephalogram (EEG), to name a few. Why did the author decide to undertake an emotional analysis on this specific work of literature as opposed to another one?
(4) In the section of the report that is devoted to the introduction, research on affective analysis that is based on deep learning is given. It is important, when discussing the study that is relevant to the topic at hand, to explain the fundamental differences between the emotion analysis system based on convolutional neural networks that was proposed by the author and these studies. This can be done when discussing the study that is relevant to the topic at hand.
(5) The author has devised a method that is superior to the traditional convolutional neural network in terms of its overall effectiveness. What is the main concept behind using the upgraded network when it comes to the analysis of the emotions included within text? It has been brought to our attention that the graphic description may benefit from some more color.
(6) Sixty percent of the dataset is used as the training set, and forty percent of the dataset is used as the test set, according to the author's description of the use of the dataset, which is not totally apparent. How to determine which sets should be used for testing and which should be used for training is the question that needs to be solved. Is it simply a matter of chance, or could there be another factor at play here?
(7) The following is the computation that needs to be done in order to determine the outcome of the experiment: Is this the result of a single experiment or a collection of findings from a number of other tests carried out separately? The piece does not offer any kind of clarification on this matter at all.

Reviewer 2 ·

Basic reporting

The high complexity of model training is one of the difficulties that needs to be addressed in this research with the present deep learning algorithm in the field of text-based emotion analysis. This is one of the concerns that needs to be solved. Word vector information is used, rather than the properties of language, which is the other fault with the model. It is necessary to find solutions to both of these problems. The author addresses the concerns that were raised earlier by utilizing a redesigned convolutional neural network model.

Experimental design

The theoretical analysis follows a strict line of reasoning, and the findings of the experiments show that it possesses a particular effect that is responsible for the enhancement of the situation.

Validity of the findings

However, in order to make enhancements to the overall quality of the post, the particulars of the language contained inside the article need to have extra optimization performed on them.

Additional comments

The following is a list of the particular problems that have arisen:
1.In order for readers to easily comprehend the aforementioned information, it is essential that the core work and contributions of the article be highlighted inside the article itself.
2.The difference between the method that is being proposed for assessing text emotions and the traditional method that is currently being used for analyzing text emotions is not described in the paper, which makes it difficult for readers to understand the subject matter.
3.The complete relevance of each individual parameter is not provided in the equations (14), and (15). Please ensure that everything is accurate and filled out to the fullest in the explanation of each parameter.
4.There are a large number of publicly accessible data sets of text emotions, such as those supplied by THUCNews, SogouCA, and other organizations. The justifications that led to the selection of the Chnsenticorp data collection for the purpose of this specific piece of writing.
5.In the experiment part, the analysis of the Recall index is very simple, and the data are presented with only a simple description; there is no attempt made to undertake a more in-depth analysis of the data.
6.The image decomposable convolution algorithm is dissected and displayed in Figure 5 along with its component elements. Due to the fact that the description of this structure does not have the required amount of specificity, it is strongly recommended that the description be revised.
7.Experiments that the author conducted with data that they created themselves are presented in sections 4 and 3, respectively. How are the data sets that an individual creates themselves collected, and what are the criteria for selecting them?
8.In the conclusion, three problems that need answers are brought to the reader's attention for consideration. You should make it clear in the section of the paper that is devoted to the conclusion whether or not the author intends to focus future research efforts on the issues that have been raised.

---

## Round 0.2 · accepted · Accept

Now, this paper can be accepted.

·

Basic reporting

No comments

Experimental design

No comments

Validity of the findings

No comments

Additional comments

An improved convolutional neural network is proposed by the author as a means of reading the author's intended feelings from the text. Positive results were observed when emotion analysis was applied to text, as shown by the experiment. The beneficial societal significance of this study is mostly attributable to the emotional analysis of college students.
Now, I personally think that the authors really revised this study seriously. I agree to accept the paper.

Reviewer 2 ·

Basic reporting

The high complexity of model training is one of the difficulties that needs to be addressed in this research with the present deep learning algorithm in the field of text-based emotion analysis. This is one of the concerns that needs to be solved. Word vector information is used, rather than the properties of language, which is the other fault with the model. It is necessary to find solutions to both of these problems. The author addresses the concerns that were raised earlier by utilizing a redesigned convolutional neural network model.

Experimental design

The theoretical analysis follows a strict line of reasoning, and the findings of the experiments show that it possesses a particular effect that is responsible for the enhancement of the situation.

Validity of the findings

However, in order to make enhancements to the overall quality of the post, the particulars of the language contained inside the article need to have extra optimization performed on them.

Additional comments

I have no questions. I think this paper can be accepted.